# *Cytinus* under the Microscope: Disclosing the Secrets of a Parasitic Plant

**DOI:** 10.3390/plants10010146

**Published:** 2021-01-12

**Authors:** Enrico Sanjust, Andrea C. Rinaldi

**Affiliations:** Department of Biomedical Sciences, University of Cagliari, Cittadella Universitaria, 09042 Monserrato, CA, Italy; sanjust@unica.it

**Keywords:** antibacterial, antimicrobials, bioactive properties, *Cytinaceae*, ethnobotany, holoparasitic, host–parasite interactions, parasitic plants, phytochemical profile, tannins

## Abstract

Well over 1% of all flowering plants are parasites, obtaining all or part of the nutrients they need from other plants. Among this extremely heterogeneous assemblage, the *Cytinaceae* form a small group of holoparasites, with *Cytinus* as the main representative genus. Despite the small number of known species and the fact that it doesn’t attack crops or plants of economic importance, *Cytinus* is paradigmatic among parasitic plants. Recent research has indeed disclosed many aspects of host–parasite interactions and reproductive biology, the latter displaying a vast array of adaptive traits to lure a range of animal pollinators. Furthermore, analysis of biological activities of extracts of the most common species of *Cytinus* has provided evidence that this plant could be a valuable source of compounds with high potential in key applicative areas, namely food production (nutraceuticals) and the development of antimicrobial therapeutics. This article offers a complete overview of our current knowledge of *Cytinus*.

## 1. Introduction

Parasitic plants are a diversified guild. Some 4750 species of angiosperms, hosted in over 290 genera, thrive deriving all or part of their nutriment from other plants through a rootlike, specialized organ called haustorium [1]. According to the latest research in the field, the parasitic lifestyle has evolved at least 12 times in flowering plants [2]. This has resulted in a bewildering morphological variety, accompanied by significant physiological and ecological adaptations, depending in large part on the ways of interactions with hosts. For example, while the majority of parasitic plants retains some photosynthetic activity, and thus are considered hemiparasites, holoparasitic plants are non-photosynthetic and thus totally depend on hosts for organic carbon [3].

The aim of this review article is to place *Cytinus* under the spotlight. This group of holoparasites grows endophytically, embedded within the host plant tissues, and flowers, the only visible part, emerge from host tissues during the reproductive period. A fair number of studies conducted in the last decade or so, have cast light on many intriguing aspects of this genus, not only related to its ecology and reproductive biology, but also to the nutritional properties, chemical composition, and biological activities of compounds extracted from some of its members.

## 2. Diversity and Distribution

*Cytinus* is a small genus in the family *Cytinaceae*. The family was traditionally grouped in the Rafflesiales, but molecular studies have supported its nesting within the Malvales, with a close relationship to *Muntingiaceae* [4,5,6]. No fossils of *Cytinaceae* exist, but estimates based on single-copy nuclear genes place the origin of *Cytinus* at 72.1 myr [7]. The genus has currently eight recognized species, with a disjunct distribution around two centres of diversity, one in the Mediterranean area and the other in southern Africa and Madagascar [8,9]. *Cytinus hypocistis* (L.) L. and *C. ruber* (Fourr.) Fritsch are two common species in the Mediterranean region and Macaronesia where they mainly parasitize roots of *Cistus* and *Halimium*, two genera of shrub plants in the family *Cistaceae*. Growing on white-flowered *Cistus* species (e.g., *C. ladanifer* L. and *C. salviifolius* L.), *Halimium*, and more rarely *Helianthemum* and *Fumana*, *C. hypocistis* is native to the entire Mediterranean basin and Canary Islands [10,11]. *C. ruber* occurs in southern Europe, from Portugal to Turkey, Bulgaria, Cyprus, Israel, Georgia, north-western Africa (Canary Islands, Morocco, Algeria, Tunisia); it parasitizes pink-flowered species of *Cistus*, like *C. creticus* L. and *C. albidus* L., with flowering extending from February to June and fruiting between August and November [12,13,14]. *C. ruber* has also been observed to parasitize *Halimium halimifolium* L. in Sardinia (Alessia Tatti, personal communication). The two species of Mediterranean *Cytinus* are easily distinguished in the field: *C. hypocistis* has bright yellow flowers while *C. ruber* has flowers with crimson or bright-red scale leaves and bracts, and an ivory-white or pale pink perianth (Figure 1 and Figure 2). Authors often mention several subspecies of both *C. hypocistis* and *C. ruber*, but the only accepted infraspecific taxon so far is *C. ruber* subsp. *canariensis* (Webb & Berthel.) Finschow ex G.Kunkel [8]. Although these Mediterranean species are considered strict root parasites, exceptional reports of *Cytinus hypocistis* emerging from the stem of host plants do exist [15], a feature that is more common in species from Madagascar (see below). Three more *Cytinus* species occur in southern Africa. While *Cytinus capensis* Marloth and *C. sanguineus* (Thunb.) Fourc. are distributed in the Cape Floristic Region, *C. visseri* Burgoyne has a more northern range, occurring in the Limpopo and Mpumalanga Provinces of South Africa and in Swaziland [16,17]. As an example of the phenology of southern hemisphere species, in *C. visseri* flowering takes place from December to May (summer to autumn), and fruits ripen the next autumn/winter from May to July [17]. Host specificity of southern African *Cytinus* species is low, these parasites being found on roots of woody shrubs of various members of *Asteraceae*, *Rosaceae*, and *Rhamnaceae*. Very little is known about the three species endemic of Madagascar (*Cytinus baronii* Baker f., *C. glandulosus* Jum., *C. malagasicus* Jum. & H. Perrier), described between 1888 and 1923; to the best of our knowledge, no modern studies on these taxa exist. A possible new species of *Cytinus* from Madagascar is depicted in [9]. Intriguingly, starting with the description of *C. baroni* by Edmund Baker in 1988 [18], it has been noticed that flowers of Madagascar *Cytinus* species often occur in sessile clusters emerging directly from the trunk of the host (see also [9]).

Besides *Cytinus*, the family *Cytinaceae* hosts two more genera, namely *Bdallophytum* (sometimes spelled *Bdallophyton*), with three neotropical species distributed from central Mexico to Costa Rica [19], and the recently described *Sanguisuga* [20], with a single species, *S. caesarea* Fern. Alonso & H. Cuadros. Being reported from northern Colombia, *Sanguisuga* is the first record of a member of *Cytinaceae* from South America, thus making a considerable range extension for the family. However, the validity of *Sanguisuga* has been recently disputed and it has been transferred to *Bdallophytum* see [2].

## 3. Life, Inside

Not surprisingly, the feature that most profoundly characterizes *Cytinus* life history and ecology, is its growing endophytically within the tissue of the host plant to obtain both water and nutrients [23]. A study of the carbon nutrition in the pair *Cistus monspeliensis* L.—*C. hypocistis*, has revealed that organic carbon is transferred from the host to the parasite mainly as sucrose, aspartic acid, asparagine, glutamic acid, and malic acid [24]. Inflorescences of *C. hypocistis* detached from the host were shown to retain the ability to fix CO_2_ to some extent (about 12% of the carbon received from the host), probably via non-photosynthetic carboxylation activity mediated by PEP carboxylase and, possibly, RuBP carboxylase, researchers hypothesized [24]. The endophytic systems of *C. hypocistis* and *C. ruber* were observed to be very similar and to spread in all host root tissues, deeper than previously thought, and to include a well-developed vascular system with xylem and functional phloem [23].

A novel three-way plant-plant-fungus association involving *C. hypocistis*, *Cistaceae* host species, and arbuscular mycorrhizal fungi, with mycorrhizae being associated with both the host and the parasite, was described by de Vega and colleagues [25]. The existence of such complex interactions, which researchers supposed to have trophic significance, with *Cytinus* acquiring nutrients both from host and mycorrhizae and thus avoiding overexploitation of host resources, has been later questioned [26,27]; if confirmed, it would represent a so far unprecedented example of a tripartite association involving an endophytic parasitic plant, its host, and mycorrhizae in natural conditions.

Although *C. hypocistis*, by far the most deeply studied species in the genus, can be defined a generalist parasite, it can be locally host-specific, selecting a smaller set of hosts from those potentially available. In a study conducted in Algarve, Portugal, over a period of six years, just three of the seven potential host species of *Cistaceae* were found to be parasitized at four independent sites [28]. Intriguingly, preferred hosts, like *H. halimifolium* and *C. monspeliensis*, were not selected on the basis of their local abundance, since other species of potential hosts were not infected even if they were more abundant than parasitized species. “Ecological niche divergence of host plants *H. halimifolium* and *C. monspeliensis* may isolate host-specific races of *C. hypocistis*, thereby potentially driving allopatric divergence in this parasitic plant,” authors remarked [28]. Supporting the possibility that interaction of *Cytinus* with its hosts is a significant force in speciation, an analysis of genetic variation of *C. hypocistis* with respect to different host species has revealed that well distinguishable genetic races of the parasite could be found on different subsets (genera and/or sections) of *Cistaceae* hosts [29]. Differentiation of the genera of *Cistaceae* took place during the Oligocene to Miocene, so much later than the presumptive appearance of *Cytinus* (see above). This fits well with the possibility that ancestral *Cistaceae* species were already infected with *Cytinus*, and that specific races (and maybe also species, like *C. ruber*) of the parasite did emerge following the radiation of the host lineages they lived with [29].

Since plastid genomes (plastomes) of parasitic plants tend to undergo significant reduction due to the loss of photosynthesis, it is of interest to investigate the structure of *Cytinus* plastome and to compare it to that of other parasitic plants. After sequencing the complete *C. hypocistis* plastome and comparing it with that of autotrophic Malvales, Roquet and colleagues [6] found an extremely reduced plastid genome, with only 16 protein-coding genes mostly encoding ribosomal proteins, tRNAs and rRNAs, and presenting no IR (inverted repeat) regions, which is approximately an 80% loss of plastid genes compared with the closest photosynthetic relatives of *Cytinus*. Current knowledge indicates that plastomes of parasitic plants vary considerably in size, especially in holoparasitic species, and that IR plays a crucial role for plastome stability and conservation, its loss being linked to a higher propensity for plastome rearrangement and decay [30].

## 4. Reproductive Biology

Pollination of *Cytinus* is another non-trivial matter. *Cytinus* flowers are large and sturdy, usually emerging from host roots at soil level (in Madagascan species flowers may protrude from the basis of host tree trunk, as mentioned above). Mediterranean species are monoecious, whereas the South African and Madagascan species are dioecious; in all cases, flowers are always unisexual. A number of studies conducted by Clara de Vega and co-workers (see below) have shed much light on the biology of reproduction of *C. hypocistis*, used as a model system. In this species, the reproductive cycle is short, with flowers blooming in April–May (spring) and ripening in mid-June–early July (late spring–early summer), and the abundantly produced tiny, dust-like seeds (approx. 25,000 per fruit) dispersed later the same months. Despite the common difficulties in assessing seed viability in holoparasitic plants, given the special requirements for germination, the application of special techniques have permitted to ascertain that in *Cytinus* fruits seeds are generally viable [31,32]. A peculiar feature is the huge number of the ovules, intermixed with an intraovaric mucillage whose role could be to prevent ovules from drying up inside the ovary in late spring or early summer in the thermophilic garigues where *C. hypocistis* and its hosts live [33]. Using a combination of field observations and experimental pollination treatments at six study sites in Spain, de Vega and colleagues [10,32,34] could gather evidence that, while foraging for nectar, ants act as efficient pollinators of *C. hypocistis*, an uncommon occurrence among angiosperms, notwithstanding the fact that ants are among the most frequent floral visitors in general. Ants accounted for over 97% of total floral visits to *C. hypocistis*, yielding a fruit set ≈80%; on the other hand, the very low number of flying insects foraging on the flowers of the parasitic plant during the study period could be due to the deterrent effect played by ants’ presence, researchers concluded [32]. Further analysis revealed that ants of at least four different species are attracted by *Cytinus* floral scent, in particular by 4-oxoisophorone, (E)-cinnamaldehyde, and (E)-cinnamyl alcohol [35]. Although pollination by ants has been hypothesized before for South African *Cytinus* [36], this has never been proved, and the reality seems to be very different, as detailed below. Ants might be the main pollinators of *Cytinus* in the Mediterranean area, but its flowers are certainly visited also by bees, at least occasionally, as demonstrated by the fact that the pollen of the parasitic plant can be traced in several types of honey [37,38,39].

The *Cytinus*-ants mutualistic affair ticks upward to a higher level of intricacy if one considers microorganisms in the mix. A couple of related studies have investigated the role of ant-transported yeasts on nectar production and composition in *C. hypocistis* flowers. Ants were found to transport both ascomycetous and basidiomycetous yeasts, with nectar concentration declining significantly with increasing yeast density in flowers [40]. Probably due to yeast metabolic activity, a decrease in sucrose and an increase in fructose and glucose content was detected in the nectar of ant-visited *C. hypocistis* flowers, these changes being correlated with the density of yeast cells in nectar [41]. Although the ecological role of nectar changes induced by ant-borne microorganisms is not clear, researchers have suggested that it might alter foraging preferences of pollinators sensitive to variations in nectar sugar constituents. In other words, “ants may therefore modify pollinator visitation not only by their presence or aggressive behavior, but also by inducing indirect chemical changes in this main food source through yeast vectoring,” [41].

But there are bigger folks than ants looking for *Cytinus* around. In South Africa, researchers have convincingly demonstrated that *C. visseri* relies on three mammal species, the striped field mouse *Rhabdomys pumilio*, the short-snouted elephant shrew *Elephantulus brachyrhynchus*, and, to a lesser extent, the pygmy mouse *Mus minutoides*, for pollination [42,43]. Intriguingly, the elephant shrew is an insectivorous species, but in this case was seen feeding eagerly on *C. visseri*’s nectar, making it the first dioecious plant species known to be mammal pollinated [43]. A single aliphatic ketone, 3-hexanone, found in the pungent scent of *C. visseri* was shown to act as a strong attractant for its primary mammal pollinator, the striped field mouse (Figure 3). More recent research has provided evidence that also *C. capensis* might depend on nocturnal ground-dwelling mammals for pollination. “Rodents access the copious nectar by pushing down individual petals along pre-formed hinges, and transfer pollen on the fur around their snouts. Insects do not visit the flowers,” researchers reported [44]. Adding to the complexity of pollination biology of *Cytinus* in the southern hemisphere, the scentless and brightly red flowers of the third South African endemic species, *C. sanguineus*, seem to be pollinated by sunbirds (*Nectariniidae*) instead [45]. Thanks to the use of trap cameras, researchers could clearly observe sunbirds visiting the ground-based flowers of *C. sanguineus*, drinking the nectar from narrow, tubular nectaries and transferring pollen on their beaks between male and female inflorescences. Insects and mammals were rarely seen visiting the flowers of *C. sanguineus*, either because they emit no scent and nectar is hard to reach by mammals in this case, thus limiting the pollination potential of both groups of animals [45].

What briefly summarized here highlights the remarkable adaptive plasticity of *Cytinus*’ reproductive biology, with floral characteristics like size, structure, and the presence or absence of volatile attractants, evolved in the various species to fit the needs of a diversified array of pollinators, either ants, mammals, or birds. The evolution of pollination systems in *Cytinaceae* is certainly a research area that deserves more attention, as adaptation to novel pollinators might be a key driving force leading to plant species differentiation [46].

If the pollination systems of *Cytinus* have received considerable attention, available information on fruit and seed dispersal is rather scant, but several hints point to zoochory as the main driver. Wood mice (*Apodemus sylvaticus*) and European rabbits (*Oryctolagus cuniculus*) were observed to feed on fleshy fruits of *C. hypocistis* in Spain, and ants removed seeds from dried fruits in the same area [10]. In the southern hemisphere, Sifakas lemurs (*Propithecus diadema*) were found to forage for inflorescences of an undetermined species of *Cytinus* in Madagascar, likely contributing to seed dispersal. Lemurs apparently use olfactory cues to locate the plants on which they feed, since in this case flowers are inconspicuously colored and grow half buried in the forest’s leaf litter, researchers concluded [47]. More recent work has permitted researchers to ascertain that some tenebrionid beetles are important seed dispersal agents of *C. hypocistis*. In the field, some 46% of captured beetles consumed *C. hypocistis* fruits, with up to 31 seeds found in individual beetle frass, and seeds were proved to be intact and viable following passage through beetles’ guts; this makes a rarely occurring case of endozoochory (dispersal via seed ingestion) carried out by insects [48]. Despite the fact that small mammals consume a larger number of fruits and thus might apparently be indicated as the prevalent seed dispersers of *C. hypocistis*, the role played by beetles should not be undervalued, researchers discussed. Indeed, since beetles often dig into the ground for shelter, “seed dispersal by beetles to below-ground microsites in proximity to host plant roots may also be vital in ensuring successful establishment of *Cytinus* seedlings in Mediterranean ecosystems,” [48].

## 5. Ethnobotanical Notes

Folklore medicine has devoted considerable attention to *Cytinus*. These parasitic plants have been traditionally used in several European settings in the treatment of dysentery, for their haemostatic and astringent properties, and for soothing the inflammations of the throat and of eyes [49,50,51]. Ethnobotanical surveys carried out in the south-central part of Sardinia confirmed that the *Cytinus* juice was used as an astringent, haemostatic, and tonic substance [52]. “The plant was known for its astringent and tonic properties: the blackish juice, squeezed and condensed, was used to make the concoctions. The astringent property was exploited in places such as Lodè, Lula and Siniscola as anti-hemorrhage, and in Sadali, Seui and Seulo as haemostatic. At Perdasdefogu, the scalp pulp was applied daily on corns and calluses as a scar-healing agent, and on the skin and inflamed mucous membranes as an astringent and anti-inflammatory remedy,” reports Atzei [53] on the popular uses of *Cytinus* in Sardinia. Knowledge about the medical properties of *Cytinus* are actually very ancient. In his commentaries on the *Materia Medica* of Dioscorides, the famous Italian physician and naturalist Pietro Andrea Mattioli describes in detail the forms of preparation and the therapeutic applications of ‘hipocisto’, “ilquale chiamano alcuni Robethro, ouero citino” [54]. In a recent article, Leonti and colleagues [55] discuss at length the possibility that at least one of the three varieties of “hipocistus” quoted by Dioscorides (red, green and white) could indeed refer to a distinct parasitic plant, *Cynomorium coccineum* L. [56].

Another common ethnobotanical use of *Cytinus* plants in Europe is as food. A recent study by Łuczaj and colleagues [57] of the taxa of wild vegetables consumed in the Adriatic islands of Croatia cited the interesting tradition of “eating the flowering shoots of the parasitic *Cytinus hypocistis* (L.) L., which is still widely known (though its practice ceased a few decades ago) on the island of Pašman”. In the Turkish province of Balıkesir, both *C. hypocistis* and *C. ruber* are consumed raw (or the juice is sucked) for their nutritious properties [58]. The consumption of *C. hypocystis* in particular has been reported before from various localities in Spain and Portugal, sometimes quoting it as a ‘famine food’ [59,60,61,62], and likely the consumption of these parasitic plants in times of food shortage was once widespread across their range of distribution. ‘Chupamieles’ and ‘meleras’ are just two of its many vernacular names (Castilian Spanish), and clearly refer to the sweet and sticky substance secreted by the inflorescences, that kids living in rural areas used to eagerly seek and suck [49].

## 6. Chemical Composition and Nutritional Characterization

Given the traditional use of *Cytinus* in popular medicine and as food, and also its striking look lightened up by brilliant colors, it is not surprising that researchers have devoted some effort to the chemical characterization of this parasitic plant. The first works on this avenue were those of Schildknecht and colleagues [63,64], who isolated the yellow pigment of *C. hypocistis* flowers and fully determined its structure. The new compound, named isoterchebin, was found to be an ellagitannin, an ester of 1,2,3-tri-*O*-galloyl-*β*-D-glucose (Figure 4). Trapain, isolated from *Trapa japonica* Flerow (*Lythraceae*), and cornus-tannin-1, extracted from *Cornus officinalis* Siebold & Zucc. (*Cornaceae*), were shown to have an identical structure to isoterchebin see [65] and references therein.

Tannins are important constituents of *Cytinus*, indeed. Methanolic extracts of *C. ruber* from Greece were shown to contain a mix of hydrolyzable tannins, namely gallotannins and ellagitannins; in some fractions, 1,2,3,6-tetragalloyl-*O*-*β*-D-glucose and 1,2,3,4,6-pentagalloyl-*O*-*β*-D-glucose were identified as the main components [66]. More recently, a High-Performance Liquid Chromatography (HPLC) and Mass Spectrometry (MS) analysis of aqueous and ethanolic extracts of *C. hypocistis* and *C. ruber* samples collected in Sardinia revealed that they contained a significant amount of tannins (up to almost 20 g/kg in the water extract of *C. hypocistis*), with β-glucogallin (1-*O*-galloyl-*β*-D-glucose) being particularly abundant [67]. The presence of pentagalloyl-*O*-*β*-D-glucose was confirmed in all extracts, reaching the concentration of 0.117 g/kg in the ethanolic extract of *C. hypocistis*, which, in general, contained a higher amount of tannins with respect to *C. ruber* [67]. A more in-depth analysis of the chemical profile of hydroethanolic extracts of *C. hypocistis* collected on *Halimium* hosts in Portugal indicated the presence of 17 phenolic compounds, with galloyl-bis-hexahydroxydiphenoyl (HHDP)-glucose, digalloyl-bis-HHDP-glucopyranose, and trigalloyl-bis-HHDP-glucose being the most concentrated [68]. As for the distribution of phenolic compounds in the various parts of the plant, petals contained the highest concentration of phenolic compounds, while nectar extracts gave the lowest levels. For a scheme of the main results of phytochemical profile studies in *Cytinus*, see Table 1.

Contributing to the understanding of the chemical characterization of *Cytinus*, and of the reasons behind its use as a traditional food, Silva and colleagues [69] have studied the nutritional composition of *C. hypocistis* from Portugal, comparing the whole plant versus its nectar. While energy (about 390 kcal/100 g), ash, and carbohydrate content were similar, protein and fat (with a net prevalence of unsaturated over saturated fatty acids) concentration was two times higher in nectar, that also contained a rather balanced mix of free sugars (fructose, glucose, sucrose, and trehalose). “Therefore, *C. hypocistis* proved to be an excellent source of nutritional compounds, which supports its use during past periods of scarcity,” the authors concluded [69]. A comparative analysis of the nutritional components of *C. hypocistis* and *C. ruber* plants collected in Sardinia has shown a substantial similarity in the nutritional profile of the two plants (Rinaldi, unpublished results). However, at variance with the data reported above for Portuguese samples, Sardinian plants contained considerably less carbohydrates (average 55.55 versus 85.95 g/100 g) and much more fiber (average 29.87 versus 2.91 g/100 g), with an obvious effect on energy content (average 320.25 versus 387.65 kcal/100 g) (Table 1). Likely, the local microenvironment and the maturation stage of the analyzed plants, coupled with genetic diversity, may well account for the observed variability.

## 7. Biological Activities

The quest for bioactivity, either in the prevention or treatment of various ailments, is the last chapter of this (ongoing) story of *Cytinus*. Early accounts have focused on the antitumoral potential of extracts of *C. hypocistis* and *C. ruber*, that displayed activity against several cell lines, above all human lung carcinoma (A549) and murine leukemia, with IC_50_ values overall ranging from 5.8 to 55 µg/mL in the case of *C. ruber* (Table 2) and hydrolysable tannins (see above) as likely key molecules [66]. Our understanding of the value of the cytotoxic/cytostatic properties of *Cytinus* extracts has been expanded by fresher evidence. Four extracts (whole plant, petals, stalks, nectar extract) of *C. hypocistis* displayed effective anti-proliferation activity against four tested tumor cell lines, namely MCF-7 (breast adenocarcinoma), NCI–H460 (non-small cell lung carcinoma), HeLa (cervical carcinoma), and HepG2 (hepatocellular carcinoma), with HeLa being the most vulnerable cell line and control non-tumour porcine liver cells (PLP2) remaining not affected even at elevated extract concentrations [68] (Table 2). On the other hand, an earlier attempt to trial *Cytinus* against the B16F10 melanoma cell line showed that both water and ethanol extracts had not significant cytotoxic effect, even at the highest tested dose (1000 μg/mL) [70]. The molecular bases of the unequal sensitivity of tumoral cell lines to *Cytinus* as tested *in vitro*, are totally ignored at this stage.

The constant and rapid raise of antimicrobial resistance rates across the world is putting national health services under increasing pressure, limiting the therapeutic options because of the inefficacy of most antibiotics currently in our deplenished arsenals. The need for new antibiotics, especially if endowed with novel mechanisms of action, is thus urgent, and plants are an obvious target in the pursuit for new antimicrobials. Several studies have explored the antimicrobial activity of *Cytinus* extracts, revealing a considerable range of susceptible bacterial strains. In a first attempt in this direction, Zucca and colleagues tested ethanolic and aqueous extracts of *C. hypocistis* against a mix of Gram-positive and -negative strains [70]. Extracts were active on all tested Gram-positive strains, including a clinical methicillin-resistant isolate of *Staphylococcus aureus*, and on the Gram-negative *Acinetobacter baumanii*. Conventional antibiotics like cloxacillin, ampicillin, and oxytetracycline, used as controls, were comparatively more active than *Cytinus* extracts, but displayed a narrower spectrum [70]. A further investigation has confirmed the activity of *C. hypocistis* and also *C. ruber* extracts on Gram-positive bacterial strains (*S. aureus*, *Staphylococcus epidermidis*, *Enterococcus faecium*), with MICs ranging from 125 to 500 μg/mL for aqueous extracts and from 31.25 to 250 μg/mL for ethanolic extracts, and also showed the suppressive activity of ethanolic extracts of both plants on biofilm formation of *S. epidermidis*, a promising finding indeed [67]. Experiments performed with synthetic pentagalloyl-*O*-*β*-D-glucose suggested that this is likely to be one of the active antimicrobial components of *Cytinus* extracts [67]. At variance with these experimental results, another couple of studies have provided evidence that extracts of *C. hypocistis* may also be active on selected Gram-negative strains like *Escherichia coli*, *Klebsiella pneumoniae*, *Pseudomonas aeruginosa*, *Morganella morganii*, and *Proteus mirabilis*, although at relatively high concentrations (in the mg/mL range) in most of the cases [68,71]. Certainly, the method and type of extraction applied in the various studies is a variable that should be considered carefully when comparing data from different works, but overall these results highlight the interest of *Cytinus* as a valuable source of potential antimicrobials, worth of more in-depth investigation.

Scouting for bioactivity among plants species from the Greek island of Crete, Fokialakis and co-workers found that methanol extracts of two subspecies of *C. hypocistis* and also *C. ruber* had significant activity against two distinct strains of *Plasmodium falciparum* (IC_50_ < 10 µg/mL), potentially attributing antimalarial activity to hydrolysable tannins [72]. In the same study, though, *C. hypocistis* and *C. ruber* were found to be depleted of activity against another pathogenic protozoan, *Leishmania donovani* [72]. A similar phytochemical survey conducted in another Mediterranean island, Sardinia, has identified in *C. hypocistis* one of the most relevant sources in the island’s flora of promising compounds with inhibitory activity against enzymes of cosmetic interest, namely elastase and tyrosinase [73]. To put screening like this into context, one should keep in mind that the cosmetic arena is an ever-growing worldwide market worth billions of dollars, and that inhibitors of elastase and tyrosinase find application, among others, as skin protectors and anti-ageing agents [74]. The extracts of Sardinian *C. hypocistis* were found to be enriched in polyphenols and flavonoids, whose activity researchers attributed the enzymatic inhibitory properties [73]. The tyrosinase inhibition activity of *C. hypocistis* and *C. ruber* extracts—even if this proved to be much lower in the latter case—could also find application in food formulations to prevent the browning for which tyrosinase (or polyphenoloxidase) is responsible [67,70]. When the anti-tyrosinase activity of the various parts of *C. hypocistis* was tested, stalks gave the best results, followed by petals, whole plant and nectar, that could inhibit just 27.6% of tyrosinase activity [68].

A large body of literature has stressed, in recent years, the value of natural antioxidants in food products, in the formulation of nutraceuticals. The antioxidant activity of extracts of *C. hypocistis* and *C. ruber*, tested with several methodologies, resulted to be particularly high in the case of ethanolic extracts, which contained significantly more phenolics than the water ones [67,70]. For both species, flavonoids accounted for only a small part of total phenolics, whereas no anthocyanins were detected. In all cases, *C. hypocistis* extracts displayed a strongest antioxidant activity than *C. ruber* extracts, so that “according to the obtained data, *C. hypocistis* can be assessed as a valuable source of antioxidant chemicals, even for food formulations”, concluded Zucca and colleagues [70]. The high antioxidant capacity of *C. hypocistis* extracts has been later confirmed by Silva and co-workers [68], who found that petals extracts display the strongest activity, in line with the fact that this part of the plant retains the higher concentration of phenolic compounds.

It is highly desirable that more biological activities be tested in the future, for *Cytinus* extracts and for plants in general. Walking this trail, Silva and colleagues [68] have recently reported that *C. hypocistis* extracts have some antidiabetic properties, expressed in terms of α-glucosidase inhibition, with stalks being the most active source, and also display anti-inflammatory activity, measured as the ability of extracts to reduce the production of nitric oxide by LPS-stimulated macrophages. Last, but not least, isoterchebin (extracted from *C. officinalis*) was shown to significantly inhibit acetylcholinesterase, butyrylcholinesterase, and β-site amyloid precursor protein cleaving enzyme 1 (BACE1), enzymes considered to be crucial targets in the treatment of Alzheimer’s Disease, which expands considerably the potential therapeutic interest of *Cytinus*-derived compounds [75].

## 8. Perspectives and Conclusions

Although *Cytinus* does not cause damage to agricultural crops, thus there are not direct, compelling economic reasons behind its study, the research that unfolded recently has undeniably shown that important lessons can be learned from this tiny, striking parasite. Interpreted as a model system, *Cytinus* investigations have provided exciting insights into the evolution of host–parasite interactions, host specificity and parasite speciation, and pollination strategy as a push towards parasite diversification. Another stimulating research avenue is indicated by the diversity of biological activities of *Cytinus* extracts. Much remains to be done in this context, especially for what concerns the isolation of most active compounds from mixtures, as the case of isoterchebin illustrates, and the evaluation of rationally designed analogs. If this approach is pursued, then other, so far unexplored areas of application could open, for example the potential of *Cytinus* as a source of natural compounds potentially active against crop pathogens or weeds [76]. Moreover, no data on antiviral activity of *Cytinus*-derived compounds still exist.

## Figures and Tables

**Figure 1 plants-10-00146-f001:**
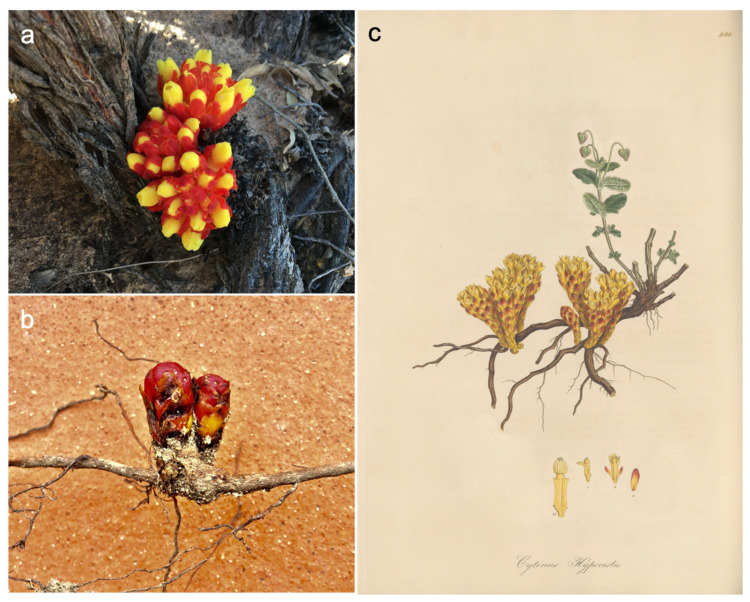
(**a**) *Cytinus hypocistis* partially opened inflorescences, at the basis of a *Halimium halimifolium*’s stem, Sardinia, April 2017, author: A.C. Rinaldi; (**b**) *C. hypocistis* emerging directly from a *Cistus salviifolius*’ root, Sardinia, March 2016, author: A.C. Rinaldi; (**c**) a classic drawing of *C. hypocistis* [21].

**Figure 2 plants-10-00146-f002:**
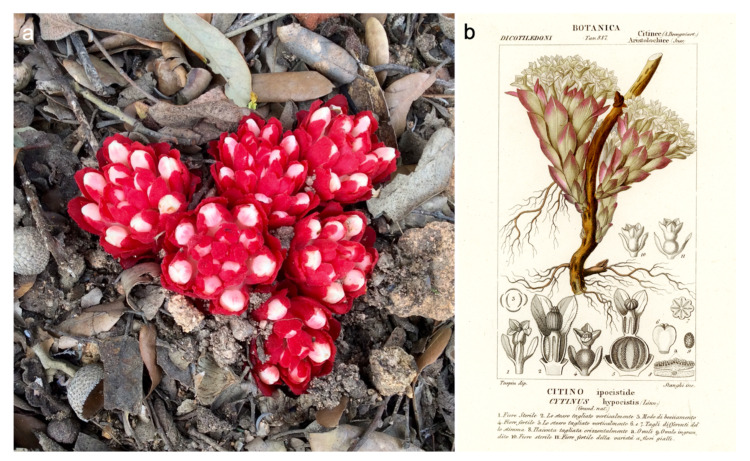
(**a**) *Cytinus ruber* inflorescences, Sardinia, April 2016, author: A.C. Rinaldi; (**b**) an 1847 drawing of *C. ruber*, clearly identifiable by its bright red and white flowers, from [22]. To note the misbranding as *C. hypocistis*: indeed, it was not until 1922 that *C. ruber* was described as a separate species.

**Figure 3 plants-10-00146-f003:**
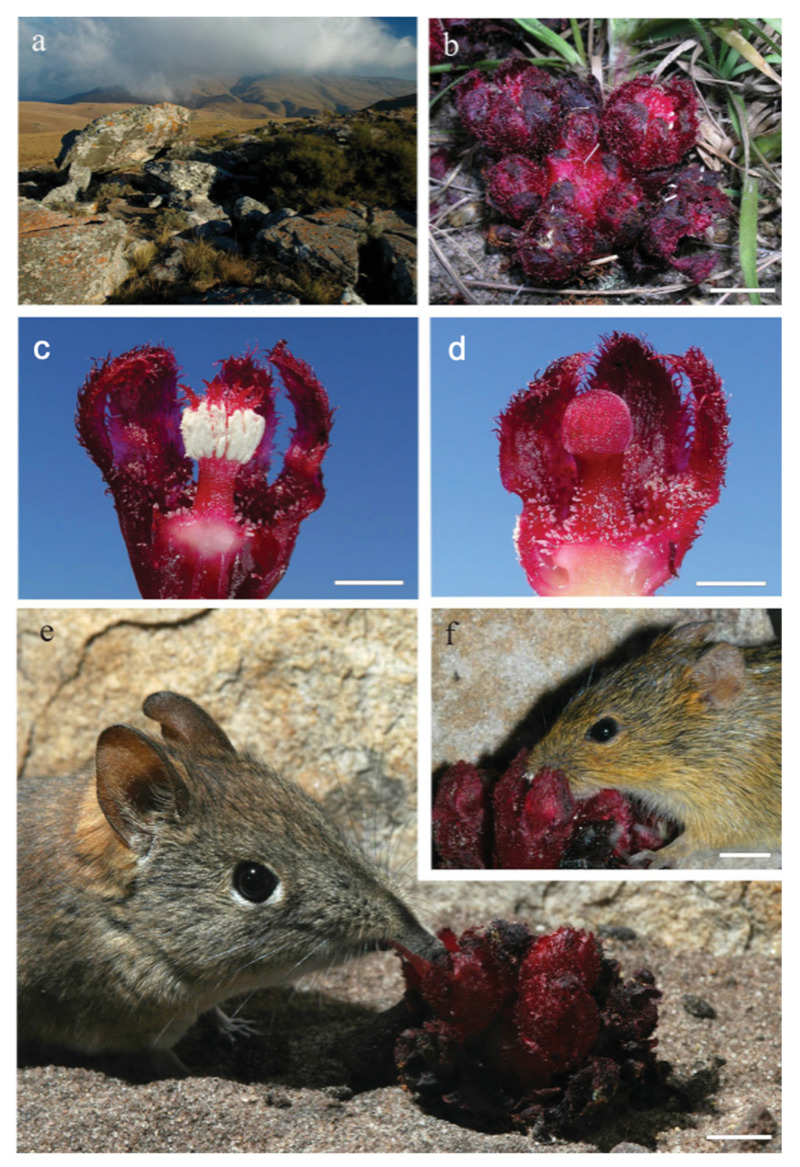
Habitat, morphology and pollinators of *Cytinus visseri*: (**a**) habitat on the summit of the Long Tom Pass, South Africa; (**b**) male inflorescence. Scale bar, 10 mm; (**c**) cross section of male flower. The nectar chamber is well visible underneath the androecium. Scale bar, 5 mm; (**d**) cross section of female flower. Scale bar, 5 mm; (**e**) a short-snouted elephant shrew *Elephantulus brachyrhynchus* feeding on nectar in *C. visseri* flowers. The tongue entering the flower is visible below the snout. Scale bar, 10 mm; (**f**) a striped field mouse *Rhabdomys pumilio* feeding on nectar in male *C. visseri* flowers. Scale bar, 10 mm. From [43]. Reproduced with permission.

**Figure 4 plants-10-00146-f004:**
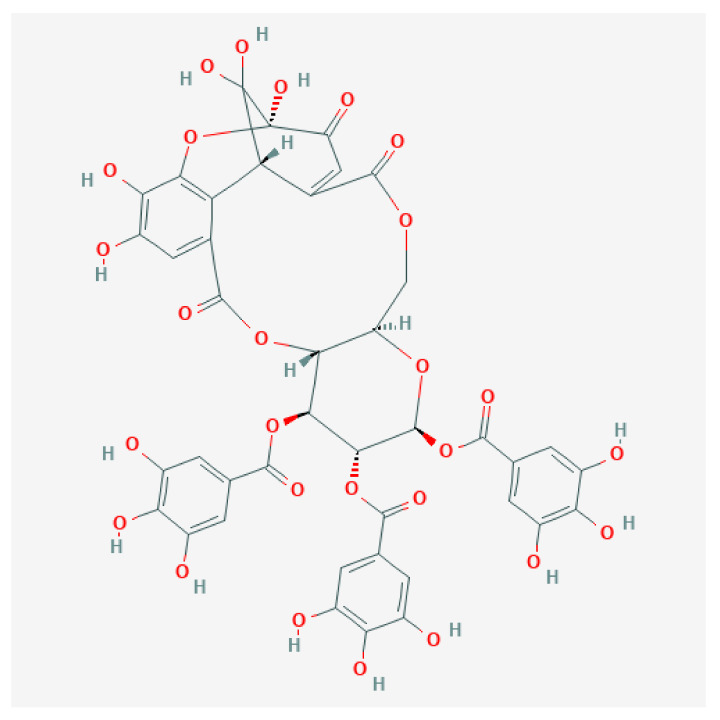
Chemical structure of isoterchebin, the yellow pigment of *Cytinus hypocistis*. In the public domain. PubChem CID: 442685, https://pubchem.ncbi.nlm.nih.gov/compound/442685.

**Table 1 plants-10-00146-t001:** Summary of results of main *Cytinus* chemical composition studies.

Species	Tannins *	Proteins	Fats	Sugars	Reference
*C. hypocistis*	isoterchebin				[63,64]
*C. hypocistis* (wp)	1-*O*-galloyl-*β*-D-glucose, di-*O*-galloyl-*β*-D-glucose, tri-*O*-galloyl-*β*-D-glucose, tetra-*O*-galloyl-*β*-D-glucose, penta-*O*-galloyl-*β*-D-glucose, tellimagradin				[67]
*C. hypocistis* (wp, pet)	galloyl-bis-HHDP-glucose, digalloyl-bis-HHDP-glucopyranose, trigalloyl-bis-HHDP-glucose				[68]
*C. hypocistis* (average wp and nect)		7.15 g/100 g	1.04 g/100 g	85.95 g/100 g	[69]
*C. hypocistis* (wp)		4.55 g/100 g	1.55 g/100 g	57.05 g/100 g	unpubl.
*C. ruber* (wp)	tetra-*O*-galloyl-*β*-D-glucose, penta-*O*-galloyl-*β*-D-glucose				[66]
*C. ruber* (wp)	1-*O*-galloyl-*β*-D-glucose, di-*O*-galloyl-*β*-D-glucose, tri-*O*-galloyl-*β*-D-glucose, tetra-*O*-galloyl-*β*-D-glucose, penta-*O*-galloyl-*β*-D-glucose				[67]
*C. ruber* (wp)		5.04 g/100 g	2.74 g/100 g	54.05 g/100 g	unpubl.

* only the most abundant and/or significant compounds are reported; nect: nectar; pet: petals; wp: whole plant; unpubl.: unpublished data (see main text for further details).

**Table 2 plants-10-00146-t002:** Spectrum of cytotoxic/cytostatic activities of *Cytinus* extracts on selected cell lines.

Species	Cell Line	IC_50_/GI_50_ (μg/mL)	Plant Part	Reference
*C. ruber* *	MDA-MB-231	37.5	wp	[66]
*C. ruber* *	BC3c	25.5	wp	[66]
*C. ruber* *	PC3	55.0	wp	[66]
*C. ruber* *	A549	5.8	wp	[66]
*C. ruber **	L1210	15.0	wp	[66]
*C. hypocistis* §	HeLa	68.0	st	[68]
*C. hypocistis* §	NCI-H460	93.0	pet	[68]
*C. hypocistis* §	MCF-7	98.0	st	[68]
*C. hypocistis* §	HepG2	77.0	st	[68]
*C. hypocistis* §	PLP2	>400	wp	[68]
*C. hypocistis*	B16F10	>1000	wp	[70]

* data refer to the activity of tannin fraction A224 [66]; § only the most active extracts are reported, see [68]; pet: petals; st: stalks; wp: whole plant.

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
