# Peer review of "Cytinus under the Microscope: Disclosing the Secrets of a Parasitic Plant"

_plants, 2021, doi:10.3390/plants10010146_

Round 1

Reviewer 1 Report

Dear Authors,

I have read with interest the manuscript entitled: “Cytinus under the microscope: disclosing the secrets of a parasitic plant” by Sanjust and Rinaldi. In this review the authors present the current knowledge of a parasitic plant, Cytinus, and the interest in studying them.  The manuscript reads well and the current knowledge on the topic is presented.

Minor comments:

Line 39: Please add if there is any information about the flowering duration

Line 45: Some information about the current status of genomic sequencing is not clearly explained even if cited and would be welcomed in this section (chloroplast genome) because this is important for future genetic diversity studies. Also any genomic particularities should be briefly mentioned in regards to others parasitic plants.

Line 60: “parasites, rare reports of Cytinus emerging from the stem of host plants do exist [e.g. 15].” this is very interesting and would deserve more than an example if possible.

Figure 1: In addition to where please indicate when as well as the authors of the pictures. Also place the figure labels within the figure to be consistent with the other figures.

Line 143-144: indicating spring and summer could be also informative, particularly for those not familiar with northern hemisphere, thanks

Line 144: please precise if they are all fertile, if any information available, thanks

Figure 3: c and d are not indicated on the figures

Line 275: HPLC and MS, please write full names

Line 519: Ref 56 indicates an additional year, please modify accordingly and check all refs

Reviewer 2 Report

In the masuscript, entitled Cytinus under the microscope: disclosing the secrets of a parasitic plant, the Authors undertook a comprehensive revision of the current state of knowledge about plants of the genus Cytinus.

In the presented work, they are precisely characterized in botanical and chemical terms, and their potential health-promoting properties are discussed. The authors used 73 literature items. The article is actually structured, it is written in scientific language, but easy to read.

All my comments and comments have been included in the attached file. After minor corrections and supplements, I believe that the article fully deserves a publication in Plants journal.

Reviewer 3 Report

The manuscript is well written, very clear, interesting results are reported and with multiple implications.
